# Expected Improvement-based Contextual Bandits

## Abstract

The expected improvement (EI) is a popular technique to handle the tradeoff between exploration and exploitation under uncertainty. However, compared to other techniques as Upper Confidence Bound (UCB) and Thompson Sampling (TS), the theoretical properties of EI have not been well studied even for non-contextual settings such as standard bandit and Bayesian optimization. In this paper, we introduce and study the EI technique as a new tool for the contextual bandits problem which is a generalization of the standard bandit. We propose two novel EI based algorithms for this problem, one when the reward function is assumed to be linear and the other when no assumption is made about the reward function other than it being bounded. With linear reward function, we demonstrate that our algorithm achieves a near-optimal regret. Further, when no assumptions are made about the form of reward, we use deep neural networks to model the reward function. We prove that this algorithm also achieves a near-optimal regret. Finally, we provide an empirical evaluation of the algorithms on both synthetic functions and various benchmark datasets. Our experiments show that our algorithms work well and consistently outperform existing approaches.

## 1 Introduction

The stochastic multi-armed bandit (Bubeck and Cesa-Bianchi, 2012; Lattimore and Szepesvári, 2020) has been extensively studied as an important model to optimize the trade-off between exploration and exploitation in sequential decision making. Among its many variants, the contextual bandit is widely used in real-world applications such as recommendation Li et al. (2010), advertising Graepel et al. (2010), robotic control Mahler et al. (2016), and healthcare Greenewald et al. (2017).

In each round of a contextual bandit, the agent observes a feature vector (the "context") for each of the $K$ arms, pulls one of them, and in return receives a scalar reward. The goal is to maximize the cumulative reward, or minimize regret (see our definition in Section 2), in a total of $T$ rounds. To do so, the agent must find a trade-off between exploration and exploitation. Currently, there are two popular techniques to solve this trade-off. The first technique is called the optimism in face of uncertainty which chooses the optimal action by maximizing upper-confidence bounds (UCB), and the second one using Thompson Sampling (TS) whose basic idea is to estimate a posterior distribution on the reward, and sample an arm that maximises a random reward drawn from this distribution. A series of work has applied both UCB and TS or their variants to explore in contextual bandits with many forms of reward functions - linear and nonlinear. In the line of UCB, there are works of (Li et al., 2010; Chu et al., 2011; Abbasi-yadkori et al., 2011) for the linear bandits, works of (Filippi et al., 2010; Valko et al., 2013) for nonlinear contextual bandits, and very recently Zhou et al. (2020), which uses neural networks to learn the reward function. In the line of TS, Agrawal and Goyal (2013); Russo and Roy (2014) are for linear bandits, Russo and Roy (2014); Kveton et al. (2020) for generalized linear functions, and Riquelme et al. (2018); Zhang et al. (2021) for nonlinear bandits using deep neural networks.

Besides UCB and TS, the expected improvement (EI) (Močkus, 1975) is one of the oldest and popular techniques to handle the tradeoff between exploration and exploitation under uncertainty. Different from UCB and TS, the EI is a greedy improvement-based heuristic that samples an action offering the greatest expected improvement over the incumbent. EI enjoys wide use due to its simplicity and ability to handle uncertainty in Bayesian optimization (Osborne, 2010; Zhan and Xing, 2020) - a

Table 1: The summary of the regret Bounds of the two popular algorithms in linear bandits.

| Algorithm | Regret Bound |
|---|---|
| Dani et al. (2008) | $\Omega(d\sqrt{T})$ (*lower bound*) |
| OFUL (Abbasi-yadkori et al., 2011) | $\mathcal{O}(d\ln(T)\sqrt{T} + \sqrt{dT\ln(T/\delta)})$ |
| LinTS (Agrawal and Goyal, 2013) | $\mathcal{O}(d^{3/2}\sqrt{T}(\ln(T) + \sqrt{\ln(T)\ln(1/\delta)}))$ |
| Our LinEI | $\mathcal{O}(d\sqrt{T\ln^2(T)\ln\frac{T}{\delta}})$ |

problem that is closely related to infinite-arm multi-arm bandits. There are also several papers Ryzhov (2016); Qin et al. (2017) using EI to study the best-arm identification problem (also known as "pure exploration") which is a finite variant of Bayesian optimization. However, for a contextual bandit which is a generalization of the standard bandit due to its reward depending on both the actions and contexts, it is yet to be seen whether EI can handle the trade-off between exploration and exploitation well. In fact, to our best knowledge, there are no EI-based algorithms even for the standard bandits in the setting of exploration and exploitation.

A key challenge of analyzing EI-based algorithms comes from its improvement function involving nonlinear, nonconvex term unlike UCB and TS. This causes the difficulty of the analysis of EI. Another challenge comes from the fact that this improvement function is mainly designed to reduce uncertainty, and thus, solely focusing on exploration. These reasons together explains why theoretical analyses of EI based methods is limited, compared to that of UCB and TS. Furthermore, these few notable exceptions typically focus on bounding the simple regret in EI based Bayesian optimisation (Bull, 2011) and best-arm identification (Ryzhov, 2016; Qin et al., 2017). To our best knowledge, none of the existing work has investigated the cumulative regret of EI in the proper bandit setting yet.

Against this background, our work proposes the first theoretical analysis of EI based methods in the classical bandit setting. In particular, we consider contextual bandits and are interested in deriving bounds on cumulative regret for our algorithms which is more suitable for contextual bandits. We note that cumulative regret bounds are stronger than the simple regret bounds and are usually more challenging to establish. In summary, our main contributions in this paper are:

- We introduce and formalize Expected Improvement as a new strategy for contextual bandits creating a parallel to UCB and TS.

- We propose two EI-based algorithms. The first algorithm (LinEI) assumes the reward function to be linear whilst the second algorithm (NeuralEI) is designed for the case when no assumption can be made about the reward function other than boundedness and we model it by a deep neural network.

- For the linear reward function, our LinEI algorithm is able to achieve $\mathcal{O}(d\sqrt{T\ln^2(T)\ln\frac{T}{\delta}})$ regret which matches the information theoretic lower bound $\Omega(d\sqrt{T})$ for this problem (up to $\ln(T)$). For the general reward function, we prove that, under standard assumptions (see section 4.1), our NeuralEI algorithm is able to achieve $\tilde{\mathcal{O}}(\tilde{d}\sqrt{T})$ regret, where $\tilde{d}$ is the "effective" dimension of a neural tangent kernel matrix and $T$ is the number of rounds.

- Finally, we provide an empirical evaluation of the algorithms on both synthetic functions and various benchmark datasets. Our experiments show that LinEI outperforms other baselines for linear bandits, and when the reward function is non-linear, NeuralEI outperforms all baselines.

## 2 PROBLEM SETTING

We consider the problem of $K$-arm contextual bandits. At time $t = 1, 2, ...$, the agent observes $K$ contextual vectors $x_{i,t} \in \mathbb{R}^d$, then selects an arm $a(t)$ and receives a reward $r_{a(t),t}$ which has a general form as follows:

$$r_{a(t),t} = h(x_{a(t),t}) + \xi_{a(t),t},$$

where $h$ is an unknown reward function satisfying $0 \leq h(x) \leq 1$ for any $x \in \mathbb{R}^d$, and $\xi_{a(t),t}$ is conditionally $R$-subGaussian for a constant $R \geq 0$, i.e., $\forall \lambda \in \mathbb{R}, \mathbb{E}[e^{\lambda \xi_{a(t),t}}|\{x_{i,t}\}_{i=1}^K] \leq \exp(\frac{\lambda^2 R^2}{2})$. In our setting, we assume that these context vectors may be chosen by an adversary in an adaptive manner after observing the arms played and their rewards up to time $t-1$. For the unknown function $h$, we consider two cases as follows:

- the reward function $h$ is linear, i.e., $h(x_{t,i}) = x_{t,i}^T \theta^*$, where $\theta^* \in \mathbb{R}^d$ are fixed but unknown parameters. Without loss of generality, we here assume that $||x_{i,t}|| \leq 1, ||\theta^*|| \leq 1$.
- the reward function $h$ is modelled by a fully connected neural network with depth $L \geq 2$ defined recursively by

$$f(x; \theta) = \sqrt{m} W_L \sigma(W_{L-1}\sigma(...\sigma(W_1 x))),$$

where $\sigma(x) := \max\{x, 0\}$ is the ReLU activation, $\theta = (\text{vec}(W_1); ...; \text{vec}(W_L)) \in \mathbb{R}^p$ is the collection of parameters of the neural network, $p = dm + m^2(L-2) + m$. Without loss of generality, we assume that the width of each hidden layer is the same (i.e., $m$) for convenience in analysis. We denote the gradient of the neural network function by $g(x; \theta) = \bigtriangledown_\theta f(x; \theta) \in \mathbb{R}^p$.

**Performance Measure.** Let $a^*(t)$ denote the optimal arm at time $t$. The objective is to minimize the cumulative regret $R(T) = \sum_{t=1}^T (x_{a^*(t),t}^T \theta^* - x_{a(t),t}^T \theta^*)$.

## 3 THE LINEI ALGORITHM FOR LINEAR BANDITS

**Prior and Posterior Distributions.** We follow the design for priors of the reward function like TS algorithm (Agrawal and Goyal, 2013). We assume that the likelihood of reward $r_{i,t}$ of each arm $i$ follows a Gaussian distribution $\mathcal{N}(x_{i,t}^\top \theta^*, v^2)$, where the variance $v^2$ will be specified later. Let $X(t) = \lambda I + \sum_{j=1}^{t-1} x_{a(j),j} x_{a(j),j}^\top$, $\hat{\theta}_t = X(t)^{-1}(\sum_{j=1}^{t-1} x_{a(j),j} r_{a(j),j})$. Then if we assume that the prior for $\theta^*$ at time $t$ is given by $\mathcal{N}(\hat{\theta}_t, v^2 X^2(t))$, then the posterior distribution of $\theta^*$ at time $t+1$ is $\mathcal{N}(\hat{\theta}_{t+1}, v^2 X^2(t+1))$ ( see the proof in Appendix A.1 in (Agrawal and Goyal, 2013)).

**Expected Improvement for Linear Contextual Bandits.** We now use this posterior distribution update to define the form of the expected improvement of each arm in contextual bandits. We denote $r_t^+ = \max_{i \in \mathcal{K}} \{x_{i,t}^T \hat{\theta}_t\}$ which is the largest mean estimate of reward among all arms at time $t$. We define the expected improvement of an arm $i$ at time $t$ as

$$\alpha_{i,t}^{EI} = \mathbb{E}_{\mu \sim \mathcal{N}(\hat{\theta}_t, v_t^2 X(t)^{-1})}[\max\{0, x_{i,t}^\top \mu - r_t^+\}], \tag{1}$$

The $\alpha_{i,t}^{EI}$ measures the potential of arm $i$ to improve upon the largest posterior reward mean at time $t$. We note that in Eq(1), $r_t^+$ plays a role as an incumbent. In Bayesian optimization, the incumbent is usually selected as the best reward value so far, or the largest reward mean so far. In our setting with contextual bandits, we choose the latter for convenient in analysis.

Further, we define $s_{i,t} = \sqrt{x_{i,t}^T X(t)^{-1} x_{i,t}}$. We see that if $\mu \sim \mathcal{N}(\hat{\theta}_t, v^2 X(t)^{-1})$ then the marginal distribution of $x_{i,t}^T \mu$ is Gaussian with mean $x_{i,t}^T \hat{\theta}_t$ and standard deviation $v s_{i,t}$. Thus, we can express the expected improvement in closed form as follows

$$\alpha_{i,t}^{EI} = (x_{i,t}^T \hat{\theta}_t - r_t^+)\Phi(\frac{x_{i,t}^\top \hat{\theta}_t - r_t^+}{v s_{i,t}}) + v s_{i,t} \phi(\frac{x_{i,t}^\top \hat{\theta}_t - r_t^+}{v s_{i,t}}), \tag{2}$$

where $\Phi(.)$ and $\phi(.)$ are the standard normal cdf and pdf function of the normal distribution respectively.

**LinEI Algorithm.** We use this expected improvement to design our algorithm. We consider two particular arms among all arms: the arm selected by the EI mechanism, denoted by $\overline{a}(t)$. Formally, $\overline{a}(t) = \text{argmax}_{i \in [K]} \alpha_{i,t}^{EI}$; and the best-so-far arm, denoted by $\tilde{a}(t)$. Formally, $\tilde{a}(t) = \text{argmax}_{i \in [K]} \{x_{i,t}^\top \hat{\theta}_t\}$.

---

**Algorithm 1** The Linear Expected Improvement Algorithm (LinEI)

---

**Input**: parameters $C_0, \beta$

1: **for** $t = 1$ to $T$ **do**
2:      Observe contexts $\{x_{i,t}\}_{i=1}^{K}$
3:      Set $\overline{a}(t) := \arg\max_{i \in \mathcal{K}} \alpha_{i,t}^{EI}$, $\tilde{a}(t) = \arg\max_{i \in \mathcal{K}}\{x_{i,t}^{\top}\hat{\theta}_t\}$
4:      **if** $\alpha_{\overline{a}(t),t}^{EI} \geq \frac{C_0}{t^{\beta}}$ **then**
5:          $a(t) = \overline{a}(t)$
6:      **else**
7:          $a(t) = \tilde{a}(t)$
8:      **end if**
9:      Play arm $a(t)$, and observe reward $r_{a(t),t}$
10:     Update $X(t+1) = \lambda I_d + \sum_{j=1}^{t} x_{a(j),j} x_{a(j),j}^{\top}$, $\hat{\theta}_{t+1} = X(t+1)^{-1}(\sum_{j=1}^{t} x_{a(j),j} r_{a(j),j})$
11: **end for**

---

Selecting arm $\overline{a}(t)$ by the EI strategy yields potentially the highest expected improvement over $\max_{i \in [K]}\{x_{i,t}^{\top}\hat{\theta}_t\}$. However, when the expected improvement (measured by $\alpha_{i,t}^{EI}$ of all arms) is small, an empirical observation is that the exploitation using $\tilde{a}(t)$ is often better than the solution $\overline{a}(t)$. To make use of this intuition, we propose a modification in the EI algorithm by introducing a threshold function $g(t)$. If $\max_{i \in [K]} \alpha_{i,t}^{EI} \geq g(t)$ then the algorithm selects $\overline{a}(t)$, otherwise selects $\tilde{a}(t)$ for the pure exploitation. We consider $g(t) = \frac{C_0}{t^{\beta}}$ which is controlled by two parameters $C_0, \beta$. Our algorithm is summarized in Algorithm 1. The threshold function is a crucial factor in achieving our optimal convergence rate for our algorithm and a relevant choice of parameters of the threshold function is discussed in the next section.

**Comparison with several related works.** Our LinEI is related to the $\epsilon$-Greedy strategy. The $\epsilon$-Greedy strategy in the standard bandit plays the best arm based on current estimates with probability $1 - \epsilon$ and otherwise explores uniformly at random. The $\epsilon$-Greedy EI algorithm in Bayesian optimization (Bull, 2011)) plays the arm with the highest EI with probability $1 - \epsilon$ and otherwise explores uniformly at random. Unlike these algorithms, our algorithm removes the exploration at random, and uses a time-varying threshold function for arm-selection instead of a coin flip. Our algorithm is also different from the TTEI algorithm (Qin et al., 2017) for the best arm identification in a non-contextual setting. TTEI algorithm is designed to allocate a sufficient number of samples to the suboptimal arms to eliminate these arms with high confidence. While this is appropriate to solve BAI but it can not be used for contextual bandit problems. This is because it is not possible to eliminate an arm completely as in contextual bandits, this arm could be the best arm in some context. See Section C in our Supplementary Material for more discussion.

### 3.1 THEORETICAL ANALYSIS

In this section, we provide the regret bound for the proposed LinEI algorithm. There are the two main challenges in our theoretical analysis. The first one is to extend several original techniques of (Bull, 2011) in Bayesian optimization which is non-contextual and noise-free setting to our setting with contexts and noisy. Different from the work of (Bull, 2011) where the reward is assumed to be in a RKHS space with a Matérn kernel, our setting assumes that the reward is a linear function of contexts and an unknown parameter $\theta^*$. The second challenge is that the EI strategy is suitable for BAI problem whose goal is to seek the arm with the best reward rather for contextual bandits which the concept "best arm" does not make sense. Solving these challenges is non-trivial. See our discussion in section C in the Supplementary Material. We achieve an upper bound for the instantaneous regret $r_t = x_{a^*(t),t}^{\top}\theta^* - x_{a(t),t}^{\top}\theta^*$ as in Lemma 9 in our Supplementary Material:

$$r_t \leq \left[\frac{\tau(\frac{\beta_t}{v_t})}{\tau(-\frac{\beta_t}{v_t})}(2\beta_t + v_t) + \left(\sqrt{2\ln(Rt^{\beta})} + 6\sqrt{C_0^{-2}d\ln(\frac{t+1}{\delta})} + 1\right)v_t\right]s_{a(t),t} + \frac{\tau(\frac{\beta_t}{v_t})}{\tau(-\frac{\beta_t}{v_t})}\frac{C_0}{t^{\beta}},$$

---

**Algorithm 2** Neural Expected Improvement Algorithm (NeuralEI)

---

**Input**: Number of rounds $T$, exploration variance $\nu$, network width $m$, regularization parameter $\lambda$ and parameters $C_0, \beta$

  1: Set $U_0 = \lambda I$
  2: **for** $t = 1$ to $T$ **do**
  3:     Set $\overline{a}(t) := \operatorname{argmax}_{i \in [K]} \alpha_{i,t}^{EI}$, $\tilde{a}(t) = \operatorname{argmax}_{i \in [K]}\{f(x_{i,t}; \theta_{t-1})\}$
  4:     **if** $\alpha_{\overline{a}(t),t}^{EI} \geq \frac{C_0}{t^\beta}$ **then**
  5:         $a(t) = \overline{a}(t)$
  6:     **else**
  7:         $a(t) = \tilde{a}(t)$
  8:     **end if**
  9:     Play arm $a(t)$, and observe reward $r_{a(t),t}$
 10:     Set $\theta_t$ to be the output of gradient descent for solving Eq(3)
 11:     $U_t = U_{t-1} + g(x_{a(t),t}; \theta_t)g^\top(x_{a(t),t}; \theta_t)/m$
 12: **end for**

---

with probability at least $1 - \delta$, where $v_t = R\sqrt{9d\ln\frac{t+1}{\delta}}$, $\beta_t = R\sqrt{d\ln(\frac{t^3}{\delta})} + 1$, $s_{i,t} = \sqrt{x_{i,t}^\top X(t)^{-1} x_{i,t}}$ and the function $\tau$ is defined as $\tau(z) = z\Phi(z) + \phi(z)$. Here, we note that parameter $v_t$ plays the role of parameter $v$ at time $t$ we discussed above. In our analysis, $v_t$ is used to eliminate the influence of $\beta_t$ so that $\frac{\beta_t}{v_t}$ is bounded as $t$ grows.

Finally, we achieve the following regret bound for our proposed algorithm with a completed proof as well as the relevant choice of parameters in Supplementary Material.

**Theorem 1.** *Given any* $\delta \in (0, 1)$. *If* $v_t = R\sqrt{9d\ln\frac{t+1}{\delta}}$, $\sqrt{d} \leq C_0 \leq d$ *and* $0.5 \leq \beta \leq 3$ *then with probability* $1 - \delta$, *the cumulative regret of the LinEI algorithm is bounded as*

$$R(T) = \mathcal{O}(d\sqrt{T\ln^2(T)\ln\frac{T}{\delta}}).$$

## 4 THE NEURALEI ALGORITHM FOR NEURAL CONTEXTUAL BANDITS

In this section, we extend our Algorithm 1 when the reward function is modelled by a fully connected neural network. Similar to the Neural Thompson Sampling approach (Zhang et al., 2021), our algorithm maintains a Gaussian distribution for each arm's reward. At time $t$, the posterior distribution of the reward of arm $i$ is updated as follows. The mean is set to the output of the neural network, denoted by $f(x_{i,t}; \theta_{t-1})$, and the variance is defined as $\sigma_{i,t}^2 = \lambda g^\top(x_{i,t}; \theta_{t-1})U_{t-1}^{-1}g(x_{i,t}; \theta_{t-1})/m$, where the matrix $U_t^{-1}$ is updated as $U_t = U_{t-1} + g(x_{a(t),t}; \theta_t)g^\top(x_{a(t),t}; \theta_t)/m$ and parameter $\theta_t$ is the solution to the following minimization problem:

$$\min_\theta L(\theta) = \sum_{i=1}^{t} [f(x_{a(i),i;\theta}) - r_{a(i),i}]^2/2 + m\lambda\|\theta - \theta_0\|_2^2/2, \tag{3}$$

where $\theta_0$ is randomly initialized network parameter. We can adapt gradient descent algorithms to solve this problem with step size $\eta$ and total number of iterations $J$ like the gradient descent algorithm of (Zhou et al., 2020).

**Expected Improvement for Neural Contextual Bandits.** We now define the form of the expected improvement in this setting. At each time step $t$, we denote $f_t^+ = \max_{i \in [K]}\{f(x_{i,t}; \theta_{t-1})\}$ which is the highest mean estimate of $f(x, \theta_{t-1})$ among all arms at time $t$. We define the expected improvement value of an arm $i$ at time $t$ as

$$\alpha_{i,t}^{EI} = \mathbb{E}_{\widetilde{f}_{i,k} \sim \mathcal{N}(f(x_{i,t}; \theta_{t-1}), \nu^2 \sigma_{i,t}^2)}[\max\{0, \widetilde{f}_{i,k} - f_t^+\}].$$

Further, the above expectation can be computed analytically as follows

$$\alpha_{i,t}^{EI} = (f(x_{i,t}; \theta_{t-1}) - f_t^+)\Phi\left(\frac{f(x_{i,t}; \theta_{t-1}) - f_t^+}{\nu\sigma_{i,t}}\right) + \nu\sigma_{i,t}\phi\left(\frac{f(x_{i,t}; \theta_{t-1}) - f_t^+}{\nu\sigma_{i,t}}\right) \tag{4}$$

Our NeuralEI algorithm is given in Algorithm 2. It starts by initializing $\theta_0 = (\text{vec}(W_1); ...; \text{vec}(W_L))$, where for each $1 \leq l \leq L - 1, W_l = (W, 0; 0, W)$, each entry of $W$ is generated independently from $N(0, 4/m)$; $W_L = (w^\top, -w^\top)$, each entry of $w$ is generated independently from $N(0, 2/m)$. NeuralEI extends our LinEI algorithm to the setting where the reward function $h$ is modelled by a fully connected neural network.

## 4.1 REGRET ANALYSIS

In this section, we provide a regret analysis of the NeuralEI algorithm. We first provide necessary background on the neural tangent kernel (NTK) theory, which plays an important role in our analysis. Following a recent line of research (Zhou et al., 2020; Zhang et al., 2021), we define the covariance between two data point $x, y \in \mathbb{R}^d$ as follows: $\tilde{H}^{(1)}(x, y) = \sigma^{(1)}(x, y) = x^\top y$, $A^{(l)}(x, y) = \begin{pmatrix} \sigma^{(l)}(x, x) & \sigma^{(l)}(x, y) \\ \sigma^{(l)}(x, y) & \sigma^{(l)}(y, y) \end{pmatrix}$, $\sigma^{l+1}(x, y) = 2\mathbb{E}_{(u,v) \sim N(0, A^{(l)}(x,y))}[\sigma(u)\sigma(v)]$, $\tilde{H}^{(l+1)}(x, y) = 2\tilde{H}^{(l)}(x, y)\mathbb{E}_{(u,v) \sim N(0, A^{(l)}(x,y))}[\sigma'(u)\sigma'(v)] + \sigma^{(l+1)}(x, y)$. Similar to (Zhou et al., 2020; Zhang et al., 2021), we assume that the number of rounds $T$ is *known* and denote the neural tangent kernel (NTK) matrix $H \in \mathbb{R}^{TK \times TK}$ based on all contextual vectors $\{x_{t,k}\}_{t \in [T], k \in [K]}$. Renumbering $\{x_{t,k}\}_{t \in [T], k \in [K]}$ as $\{x_i\}_{i=1,...,TK}$, then each entry $H_{ij}$ is defined as

$$H_{ij} = (\tilde{H}^{(L)}(x_i, x_j) + \sigma^{(L)}(x_i, x_j))/2, \tag{5}$$

for all $i, j \in [TK]$. Based on the above definition, we impose the following assumption on the contexts generated by the adversary and the corresponding NTK matrix $H$.

**Assumption 1.** *Let $H$ be defined in Eq(5). There exists $\lambda_0 > 0$ such that $H \geq \lambda_0 I$. In addition, for any $t \in [T], k \in [K]$, $||x_{t,k}||_2 = 1$ and $[x_{t,k}]_j = [x_{t,k}]_{j+d/2}$.*

**Remark 1.** Compared to Algorithm 1 for linear bandits, our Algorithm 2 needs an additional Assumption 1 to guarantee the convergence. The assumption that the NTK matrix is positive definite has been considered in prior work on NTK which is a mild condition and also imposed in other related works (Arora et al., 2019; Du et al., 2019; Zhou et al., 2020; Zhang et al., 2021). The assumption on contexts ensures that $f(x_{i,t}; \theta_0) = 0$ for any $i \in [K], t \in [T]$.

The NTK technique builds a connection between deep neural networks and kernel methods. It enables us to adapt some complexity measures for kernel methods to describe the complexity of the neural network through the notation of the effective dimensions as defined in (Zhou et al., 2020; Zhang et al., 2021). The effective dimension $\tilde{d}$ of matrix $H$ with regularization parameter $\lambda$ is defined as $\tilde{d} = \frac{\log \det(I + H/\lambda)}{\log(1 + TK/\lambda)}$.

Using these notations, we are now ready to present the second main result of the paper. Let $a^*(t) = \text{argmax}_{i \in [K]} \mathbb{E}[r_{i,t}]$ be the optimal action at round $t$ that maximizes the expected reward, we define the *expected cumulative regret* after $T$ iterations as $\overline{R}(T) = \mathbb{E}[\sum_{t=1}^{T} (r_{a^*(t),t} - r_{a(t),t})]$. Then, we achieve the following upper regret bound for our Algorithm 2 by combining our EI techniques for LinEI with NTK techniques. A completed proof is provided in Supplementary Material.

**Theorem 2.** *Under Assumption 1, set the parameters in Algorithm 2 as $\lambda = 1 + 1/T$, $\nu = B + R\sqrt{\tilde{d}\log(1 + TK/\lambda) + 2 + 2\log(1/\delta)}$, where $B = \max\{1, \sqrt{2h^\top H^{-1}h}\}$ with $h = (h(x_1), ..., h(x_{TK}))^\top$. If $\sqrt{\tilde{d}} \leq C_0 \leq \tilde{d}$, $\beta \geq 2$, and the network width $m$ satisfies $m \geq \text{poly}(\gamma, T, K, L, \log(1/\delta))$, then with probability at least $1 - \delta$, the regret of Algorithm 2 is bounded as*

$$\overline{R}(T) \leq \mathcal{O}(\tilde{d}\sqrt{\beta \log(1 + TK)\log(T)T}).$$

**Remark 2.** The regret bound depends on the parameter $\beta$. The best choice is $\beta = 2$ that tightens the regret. Theorem 2 implies the regret of NeuralEI is on the order of $\tilde{\mathcal{O}}(\tilde{d}\sqrt{T})$. Similar to previous results (Zhou et al., 2020; Zhang et al., 2021), our results require a large value of $m$. This is rooted in the current deep learning theory based on the neural tangent kernel.

## 5   RELATED WORKS AND DISCUSSION

Given the vast literature on bandit algorithms, we restrict our review to linear bandits and neural contextual bandits.

**Linear Contextual Bandit.**   A lower bound of $\Omega(d\sqrt{T})$ for linear bandits was given by Dani et al. (2008), when the number of arms is allowed to be infinite. (Abbasi-yadkori et al., 2011) analyze a UCB-style algorithm and provide a regret upper bound $\mathcal{O}(d\log(T)\sqrt{T} + \sqrt{dT\log(T/\delta)})$. When the number of arms $K$ is finite, (Chu et al., 2011) achieve a regret bound of $\mathcal{O}(\sqrt{Td\ln^3(KT\ln T)/\delta})$ with probability at least $1 - \delta$. (Bubeck et al., 2012) provides an algorithm based on exponential weights, with regret of order $\mathcal{O}(d\sqrt{T\log K})$. These algorithms may not be effective when the number of arms $K$ is large. For example, when $K$ is exponential in $d$, the regret bound of (Chu et al., 2011) would become $\mathcal{O}(d^2\sqrt{T})$ showing a quadratic growth in $d$. We note that although our algorithm is a Bayesian approach, our regret bounds will hold irrespective of whether or not the actual reward distribution matches the Gaussian likelihood function. Thus, our bounds for EI algorithm are directly comparable to the UCB family of algorithms. This is also mentioned in Agrawal and Goyal (2013).

The Thompson Sampling algorithm (Agrawal and Goyal, 2013) and an alternative given by Abeille and Lazaric (2017) often bear an additional $\sqrt{d}$ in the regret bound compared to UCB based bounds. See our Table 1. Very recently, Kim et al. (2021) improved this regret bound of TS by integrating a doubly robust estimator with TS. However, this work requires additional significant computations and their setting is restrictive in the sense that contexts need to be independent. Our EI-based algorithm achieves the same regret bound order as in Kim et al. (2021), however, it caters to a more general setting where the contexts may be controlled by an adaptive adversary and therefore may not be independent.

Another approach for linear bandits is the Information Directed Sampling (IDS) which was introduced by Russo and Van Roy (2014). It provides an action-selection mechanism by minimizing the information ratio between the squared expected regret and the mutual information between optimal action and the next observation over all action sampling distributions. IDS obtains a performance improvement over TS and UCB algorithms in some cases, but has heavy sampling requirements. It has been shown in their experiments that IDS requires significantly more compute time than Thompson sampling and UCB algorithms. Recently, (Baek and Farias, 2021) provided a modification of the arm scoring rule of IDS to reduce computations. However, both Russo and Van Roy (2014) and (Baek and Farias, 2021) only provide the bounds on *expected regret*. In contrast, our work provides regret bounds in terms of cumulative regret which is tighter than expected regret.

**Neural Contextual Bandit.**   Neural contextual bandits are becoming attractive due to the current advancement in optimization and generalization of deep neural networks (Arora et al., 2019; Du et al., 2019). While Neural contextual bandits have been considered in both popular techniques UCB (Zhou et al., 2020) and TS (Zhang et al., 2021), we consider this problem in a new setting using the EI technique. A recent work of Xu et al. (2020) combines the deep learning and the representation learning to improve the computational efficiency of the previous works, however, it considers a weaker version of the general reward function that is a linear function of deep network based extracted features.

Due to the space limitation, we added more discussion on related works in Section B in the Supplementary Material.

## 6   EXPERIMENTS

### 6.1   LINEAR BANDITS

In this subsection, we assess the performance of our LinEI algorithm on several benchmark dataset including `covertype`, `magic`, `avila`, `dry bean`, `statlog`, `letter`, `pendigits`, all from UCI (Dua and Graff, 2017). We compare the LinEI with methods designed for linear bandits including: LinTS (Agrawal and Goyal, 2013), LinUCB (Abbasi-yadkori et al., 2011), Linear Epsilon Greedy for the linear reward, LinIDS Russo and Van Roy (2014) for linear bandits. To transform these classification problems into multi-armed bandits, we adapt the disjoint models to build a context

Table 2: Characteristics of benchmark datasets used in Section 5.2.

| Dataset | letter | pendigits | covertype | avila | magic | dry bean | statlog |
|---|---|---|---|---|---|---|---|
| Classes ($K$) | 26 | 10 | 7 | 12 | 2 | 7 | 7 |
| Feature Dimension | 17 | 16 | 54 | 10 | 10 | 16 | 8 |
| Dataset size | 20000 | 10992 | 581012 | 20867 | 19020 | 13611 | 58000 |

feature vector for each arm: given an input feature $x \in \mathbb{R}^d$ of a $k$-class classification problem, we build the context feature vector with dimension $kd$ as $x_1 = (x; 0; ...; 0), x_2 = (0; x; ...; 0), ..., x_k = (0; 0...; x)$. The algorithm generates a set of predicted reward and pulls the greedy arm. For these classification problems, if the algorithm selects a correct class by pulling the corresponding arm, it will receive a reward as 1, otherwise 0. The cumulative regret over time horizon $T$ is measured by the total mistakes made by the algorithm.

We set the time horizon of our algorithm to 10000 for all data sets. In the experiments, we shuffle all datasets randomly. For $(\lambda, \nu)$ used in LinUCB and LinTS and our algorithm, we set $\lambda = 1$ following previous works and do a grid search of $\nu \in \{1, 0.1, 0.01\}$ to select the parameter with the best performance. All experiments are repeated 10 times, and the average with standard error are reported. For LinIDS, we use the number of samples $M = 100$. For Linear Epsilon Greedy, we use $\epsilon = 0.1$. For our LinEI algorithm, we can choose any value $C_0 \in [\sqrt{d}, d]$ and $\beta \in [0.5, 3]$. For LinEI, we set $C_0 = \sqrt{d}$ and $\beta = 2$.

Figure 1 shows the total regret of all algorithms for datasets `bean`, `covertype` and `statlog`. The Linear Epsilon Greedy performs the worst. This implies that the random exploration is not as effective as other methods. While LinUCB, LinTS and LinIDS are competitive, all these methods are significantly outperformed by the proposed algorithm. The arm-selection of our LinEI bases on two strategies: expected improvement and greedy strategy. Compared to Linear Epsilon Greedy, our greedy strategy is similar. It confirms that the exploration of the expected improvement is effective. This suggests that using the expected improvement strategy is efficient in linear bandits. Due to space limit, the additional results on `magic`, `pendigits`, `letter`, and `avila` are shown in Section A of Supplementary Material.

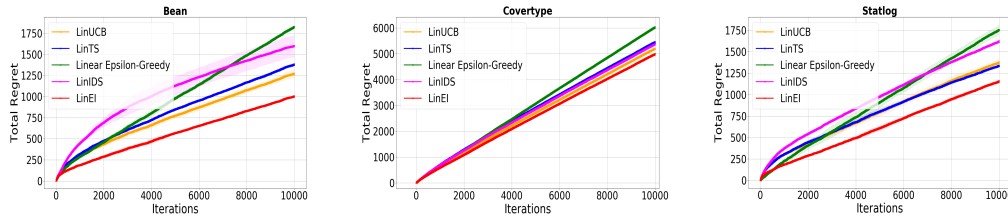

Figure 1: Comparison of our proposed LinEI and baseline algorithms in linear bandits.

## 6.2 NEURAL BANDITS

We compare the proposed NeuralEI with baselines including: LinUCB (Abbasi-yadkori et al., 2011), our LinEI for linear bandits problem, Neural Epsilon Greedy, NeuralUCB (Zhou et al., 2020), NeuralTS (Zhang et al., 2021). We do the same classification problems as the experiments in subsection Linear Bandits. For methods using the neural network, we use one-hidden layer neural networks with 100 neurons to model the reward function. During posterior updating, gradient descent is run for 100 iterations with learning rate 0.001. For Neural UCB/Thompson Sampling and Neural EI, we use a grid search on $\lambda \in \{1, 10^1, 10^{-2}, 10^{-3}\}$ and $\nu \in \{10^{-1}, 10^{-2}, 10^{-3}, 10^{-4}, 10^{-5}\}$. We consider our algorithm on both synthetic datasets and real-world datasets.

**Synthetic Datasets.** In these experiments, we use contextual bandits, we use contextual bandits with dimension $d = 10$ and $K = 5$ actions. The context vectors $\{x_{1,1}, ..., x_{T,K}\}$ are chosen

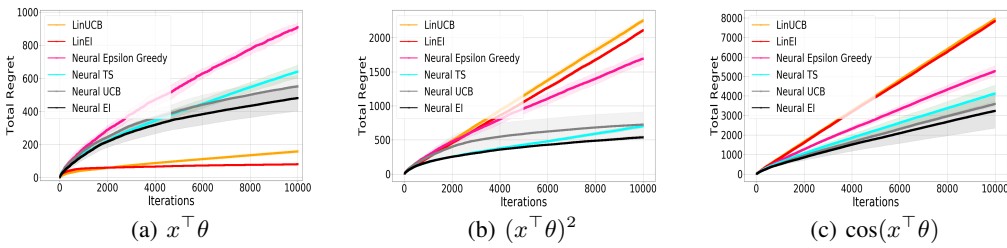

Figure 2: Comparison of NeuralEI and baseline algorithms on synthetic reward functions.

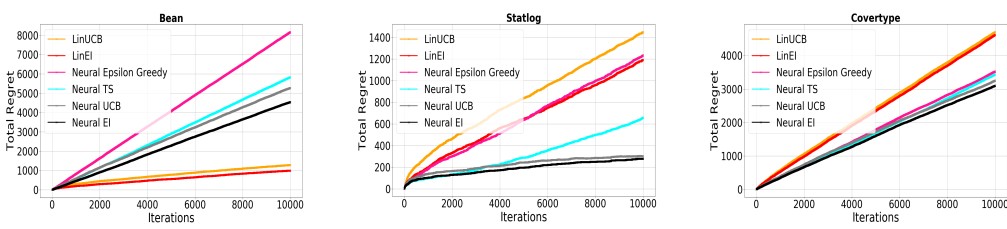

Figure 3: Comparison of NeuralEI and baseline algorithms on real-world datasets.

uniformly at random from the unit ball. The reward function $r$ is one of the following synthetic functions $r_1(x) = (x^\top\theta)^2$, $r_2(x) = (x^\top\theta)^2$, and $r_3(x) = \cos(3x^\top\theta)$.

**Real-world Datasets.** Similar to the subsection Linear Bandit, we build the context feature vector with dimension $kd$ as $x_1 = (x; 0; ...; 0), x_2 = (0; x; ...; 0), ..., x_k = (0; 0...; x)$. We also estimate our algorithm on datasets `bean`, `covertype` and `statlog`.

Figures 2 and 3 show our results in the case of the neural bandits problem. In Figure 2, if the reward function is linear, the LinEI outperforms all other neural-based methods because it is designed for the linear bandits. Otherwise, if reward functions are non-linear, LinEI and LinUCB fail to learn them for nearly all tasks due to the nonlinearity of reward functions $h$. Neural-based methods perform better because they can capture the nonlinearity of the underlying reward function. In real-datasets, while neural-based methods outperform LinEI and LinUCB for datasets `covertype` and `statlog`, these methods are not sample-efficient for learning the reward function of dataset `bean`. Perhaps, the reward function for dataset `bean` is linear. However, in all cases, our NeuralEI algorithm performs better than other neural-based methods. This suggests that using the expected improvement strategy is effective in both linear bandits and neural contextual bandits.

## 7 CONCLUSION

We introduced and formalized Expected Improvement as a new strategy for contextual bandits. We proposed two EI-based algorithms and analyzed them theoretically. The first algorithm assumes the reward function to be linear whilst the second algorithm is designed for the case when the reward function is general and can be modelled by a deep neural network. In particular, our LinEI algorithm for linear bandits achieves a near-optimal regret bound and improves the bounds of OFUL and LinTS algorithms. Our promising empirical results on both synthetic and real-world datasets suggested that our algorithms work well in practice compared to other approaches. We believe our work would be useful for further improvements and extensions. There are several interesting open questions. For example, we can extend our work to non-Gaussian distributions to model the unknown reward model parameters and derive regret bounds as long as concentration inequalities can be established. In another direction, we have used NTK in our work, but the NTK theory assumes overparameterised networks and an extension to narrow networks while maintaining the generalization of the reward function is an interesting open problem.

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
