# OpenReview forum: "Expected Improvement-based Contextual Bandits"
_ICLR.cc/2022/Conference — ICLR 2022 Submitted_

### Official Review · Reviewer_co1L · 2021-10-21

**Correctness:** 4
**Technical Novelty And Significance:** 3
**Empirical Novelty And Significance:** 2
**Recommendation:** 6
**Confidence:** 4

**Main Review:**

Generally speaking, this work is well-organized and it is interesting in the sense that it is the first the analyze EI-based algorithms for RM in linear bandits. Besides, it shows that the theoretical and numerical performance of the two proposed algorithms.  My major comments are as follows:
1. The design of the algorithm and the analytical methods seem to be quite standard. Despite the existing discussions, it would be better if the author(s) can further highlight the difference between the proposed algorithm in Qin et al. (2017) and also the challenge to derive the bounds. For example, although the analysis of linEI is a bit different from that of linUCB and linTS, what is the key challenge considering the existing EI analysis for BAI?
2. The lower bound by Dani et al. (2008) should also be placed in Table 1. Moreover, I suggest to compare to the bounds in "The End of Optimism? An Asymptotic Analysis of Finite-Armed Linear Bandits" (http://proceedings.mlr.press/v54/lattimore17a/lattimore17a.pdf).
3. Since the linear bandit setting is a generalization of the standard bandit setting, the authors can consider to also compare the linear algorithm to 'non-linear algorithms' in a standard setting. Both theoretical and numerical comparisons should be possible.
4. The numerical experiments do not consider a traditional bandit problem. Although it is good to generalize the application of bandit algorithms, what is the numerical comparison in a standard bandit problem, e.g. those in Qin et al. (2017)?

**Summary Of The Paper:**

This work proposes two Expected Improvement (EI)-based algorithms to solve the regret minimization problem for the linear and neural bandits. It provides regret analysis for both algorithms and also comparison to some existing algorithms.

**Summary Of The Review:**

The paper is okay but the contribution to the study of bandit problems does not seem to be significant. I think it would be better if the authors can highlight their contributions pertaining to what I suggested above.


=======================
Thanks for response from the authors. I am increasing my score from 5 to 6.

---

> ### Author Response · Authors · 2021-11-23
> **Response to Reviewer co1L**
>
> We sincerely thank the reviewer for the comments and suggestions. Please find our responses in the following.
>
> **On the claim that the analysis of linEI is a bit different from that of linUCB and linTS**.
> We would like to highlight that the analysis of LinEI is different from those of linUCB and linTS. Only Lemma 1 and Lemma 3 are borrowed from the analysis of linUCB and linTS, but the analysis of Lemmas 4, 5, and 6 are exclusive to linEI using the properties of the EI strategy and the choice of threshold function from our proposed algorithm. These are quite different from the techniques of the optimism principle (linUCB) and Thompson sampling.
>
> We also discuss the difference between our work with BAI and the work in Qin et al (2017). Please see our explanation in the comments to all reviewers.
>
> ** I suggest comparing to the bounds in "The End of Optimism? An Asymptotic Analysis of Finite-Armed Linear Bandits". **
> Thank you for suggesting this interesting work (Lattimore et al. (2017)) which we call [3] for simplicity. We have added the following discussion of [3] in the Related Works section.
>
> [3] analyzes the asymptotic regret for linear bandits. While showing that both optimism principle and Thompson sampling cannot be close to asymptotically optimal in the setting of linear bandits, [3] proposed a new allocation matching-based algorithm that obtains an optimal asymptotic regret. In this paper, we analyze the worst-case regret of EI-based algorithms for the linear contextual bandits and neural contextual bandits which are more general than the linear bandit considered in [3]. To our knowledge, the analysis of the optimal asymptotic regret for contextual bandits as we consider  (where the context may be controlled by an adversary) is still an open problem.
>
> **The contribution to the study of bandit problems does not seem to be significant.**  In our knowledge, most of the current works including theoretical works and imperial works for Contextual Bandits or further for Reinforcement Learning uses either the optimism principle or Thompson sampling. In this paper, we provide a different technique, the first step to integrating EI with contextual bandits and thus with reinforcement learning which is a generalization problem of contextual bandits.  Our results have theoretical guarantees.  Moreover, as the reviewer can see in our numerical comparisons,  our EI-based methods outperform all baselines (including IDS, an Information-Directed Sampling method) on seven real-world datasets as well as synthetic functions. Therefore, it is also potential to improve the asymptotic regret compared to UCB and Thompson sampling. By all these, we believe that our results are significant.
>
> **Since the linear bandit setting is a generalization of the standard bandit setting, the authors can consider to also compare the linear algorithm to 'non-linear algorithms' in a standard-setting. Both theoretical and numerical comparisons should be possible.**
>
> We apologize for not being able to answer this question as it is not clear to us. If possible, could you make clear more this question. We are always ready to do more experiments.

---

### Official Review · Reviewer_ccLd · 2021-11-02

**Correctness:** 4
**Technical Novelty And Significance:** 4
**Empirical Novelty And Significance:** 3
**Recommendation:** 6
**Confidence:** 3

**Details Of Ethics Concerns:**

no concern

**Main Review:**

This paper shows an innovative algorithm using expectation improvement and the corresponding regret analysis.  It improves the upper bound from UCB by \sqrt{T}.   Although it is debatable whether the upper bound is improved from Thompson sampling since the proposed algorithm does not use any randomization, it has a better upper bound than Thompson sampling by \sqrt{dT}.   Using expected improvement for exploration tools seems to be a good idea.

The regret analysis is interesting and innovative.

The authors may give a more intuitive explanation of how \sqrt{d} is shaved compared to Thompson sampling.  It seems like computing expected improvement instead of randomly drawing the estimator from the distribution, a new regret decomposition is possible, and thus \beta_t and v_t appear in an additive manner instead of multiplicative.

Experiments seem to show the regret grows linearly.  Have you tried a larger T?



**Summary Of The Paper:**

The authors study regret analysis of the expected improvement (EI), a popular but theoretically understudied technique to handle the tradeoff between exploration and exploitation in bandits.  They propose two novel EI-based algorithms for this problem, one for linear payoff and for deep neural networks. With a linear reward function, we demonstrate that our algorithm achieves a near-optimal regret. In particular, our regret bound reduces a factor of \sqrt{log T} from UCB. They also present numerical studies using the proposed algorithm.

**Summary Of The Review:**

This paper is well written and shows novel contextual bandit algorithms using expected improvement with new regret analysis.   This could be a nice contribution to the bandit community.

---

> ### Author Response · Authors · 2021-11-23
> **Response to Reviewer  ccLd**
>
> We sincerely thank the reviewer for the very positive comments and suggestions. Please find our responses in the following.
>
> **The authors may give a more intuitive explanation of how $\sqrt{d}$ is saved compared to Thompson sampling. It seems like computing expected improvement instead of randomly drawing the estimator from the distribution, a new regret decomposition is possible, and thus $\beta_t$ and $v_t$ appear in an additive manner instead of multiplicative.**
>
> In our understanding,  the reason why Thompson sampling has a factor $\sqrt{d}$ compared to LinUCB and our LinEI comes from the concentration inequality to bound the gap between $\theta_{i,t}$ and the mean value of the distribution where $\theta_{i,t}$ is sampled. Please see Definition 4 in the work of Agrawal et al.,(2014) for details. Bounding this gap is needed for Thompson sampling because its analysis needs to use $\theta_{i,t}$. In our analysis, we do not use the sampling and we do not need to use this concentration inequality.  Thus, we can save $\sqrt{d}$ compared to Thompson sampling. Thank you for your interesting question!
>
> **Experiments seem to show the regret grows linearly. Have you tried a larger $T$?**
> Linear growth of regret (especially up to small to medium iterations), is empirically seen in many existing works e.g., Zhou et al., 2020; Zhang et al., 2021. We think that it depends on the reward function. For example, in our experiments for synthetic functions in Figures 2(a) and 2(b), the regret grows only sub-linearly. We have also provided some additional experiments in the supplementary material, where we can see the regret of LinEI growing quite sub-linearly for the \emph{letter} dataset. After the reviews, we have increased the number of iterations from 10000 to 15000 for the \emph{letter} dataset and 30000 for the \emph{stalog} dataset.  Please see Figure 5 in our Supplimentary Material.

---

### Official Review · Reviewer_TJpE · 2021-11-07

**Correctness:** 3
**Technical Novelty And Significance:** 3
**Empirical Novelty And Significance:** 2
**Recommendation:** 6
**Confidence:** 4

**Main Review:**

Strengths:
- The paper introduces and formalizes Expected Improvement as a new strategy for contextual bandits. The EI idea has been around in Bayesian optimization but has not been adapted to contextual bandits.
- The EI is adapted to show regret bounds in linear bandits and neural bandits, both resulting in sublinear regret bounds.
- The empirical experiments show that LinEI outperforms other baselines for linear bandits and NeuralEI outperforms all baselines under the non-linear reward functions the paper considers.

 Weaknesses:
- In addition to regret analysis in linear bandits for LinEI, the regret analysis for NeuralEI is rather straightforward using the existing NTK techniques used in Zhou et al., 2020; Zhang et al., 2021. I am not sure what the analysis of NeuralEI adds to the contribution given by LinEI.
- The claim on the neural version that "no assumption is made about the reward function other than it being bounded" isn't necessarily true. It needs to satisfy certain conditions besides boundedness. See the discussions in Zhang et al., 2021.

**Summary Of The Paper:**

The paper introduces and studies the expected improvement (EI) technique as a way to balance exploration and exploitation for the contextual bandit problem. The authors propose two EI-based algorithms for linear bandits and for neural bandits for a general class of reward functions. The paper presents regret bounds for both methods and shows the experimental results to support their theoretical claims.

**Summary Of The Review:**

For the reasons above, I am leaning towards weak acceptance.

========== Post-reponses =================
Thanks to the authors for the responses, I am staying with the current score.

---

> ### Author Response · Authors · 2021-11-23
> **Response to Reviewer TJpE**
>
> We sincerely thank the reviewer for the positive comments. Please find our responses in the following.
>
> **In addition to regret analysis in linear bandits for LinEI, the regret analysis for NeuralEI is rather straightforward using the existing NTK techniques used in Zhou et al., 2020; Zhang et al., 2021. I am not sure what the analysis of NeuralEI adds to the contribution given by LinEI.**
>
>  While at a high level it is true that the regret analysis for NeuralEI combines the proof techniques of LinEI and the existing NTK techniques (Zhou et al., 2020; Zhang et al., 2021), we still needed some additional proof techniques for NeuralEI. Due to the problem of typing math formulas in OpenReview please see the paragraph "Due to the problem of typing math formulas in OpenReview please see Section B for details in Supplementary Material." on page 21 in our Supplementary Material for details. We apologize for this inconvenience!
>
> **The claim on the neural version that "no assumption is made about the reward function other than it being bounded" isn't necessarily true. It needs to satisfy certain conditions besides boundedness. See the discussions in Zhang et al., 2021.**
> Thank you for pointing this out! We have now corrected this claim/assumption.

---

### Official Review · Reviewer_vUVo · 2021-11-08

**Correctness:** 2
**Technical Novelty And Significance:** 2
**Empirical Novelty And Significance:** 3
**Recommendation:** 3
**Confidence:** 4

**Main Review:**

Major comments:
1. There are some existing papers that provide regret bounds for EI-type algorithms, for example, [1] and [2]. I think citing and comparing with these papers is necessary.
2. LinEI and NeuralEI modify the original EI by using a threshold function, see Line 4 in Algorithm 1 or Algorithm 2. Basically, LinEI and NeuralEI only sample the arm with the largest EI value if and only if the largest EI value is larger than this threshold function; otherwise LinEI and NerualEI sample the arm with the largest posterior reward mean. Intuitively speaking, I think if the largest EI value is small, the arm with the largest EI value is (almost) the same as the arm with the largest posterior reward mean. Hence, I am wondering whether introducing this threshold function is necessary. Do regret bounds still hold without this modification? If not, can the authors elaborate more on the technical difficulties as well as how this modification solves the issues?
3. Inflating the posterior variance is somewhat necessary for randomized algorithms (e.g., Thompson sampling) in order to show the optimism. I am wondering whether inflating the posterior variance is necessary for deterministic algorithms LinEI and NeuralEI.
4. In my opinion, the uniform constant $C'$ in the proof of Lemma 10 depends on $\delta$ (for example, consider $t=1$). If so, the uniform constant $C$ also depends on $\delta$ and then the final regret bound could have worse dependence on $\delta$.
5. I am wondering where Lemma 3 is used in the proof.
6. Both $\ln$ and $\log$ are used in the paper. I think it is consistent to use only one of them.

There are so many typos in this paper. For example,
1. In the second sentence of Section 2, it should be $x_{i,t}$.
2. In the paragraph Performance Measure, the summation should from $t=1$ instead of $i=1$.
3. In the paragraph Prior and Posterior Distributions, there are three normal distribution. I think none of the variance expressions is correct.
4. In Equation (2), it should be $s_{i,t}$ instead of $s_i(t)$.
4. There are so many missing brackets in the supplement material.


[1] Nguyen, V., Gupta, S., Rana, S., Li, C., Venkatesh, S.. (2017). Regret for Expected Improvement over the Best-Observed Value and Stopping Condition.
[2] Ziyu Wang, Nando de Fretias. (2014). Theoretical Analysis of Bayesian Optimisation with Unknown Gaussian Process Hyper-Parameters.


**Summary Of The Paper:**

The paper extends and modifies the expected improvement (EI) algorithm for contextual bandits. Specifically, LinEI and NeuralEI are proposed for the linear reward function and general reward function (under some assumptions), respectively. The paper claims that LinEI and NeuralEI achieve the state-of-the-art regret bounds and they work well on both synthetic functions and benchmark datasets.

**Summary Of The Review:**

As mentioned above, some important references are not cited. To show the technical novelty, detailed comparison with these references is needed. In addition, since the uniform constants in the proof do not have clear dependence on the important parameter $\delta$ and there are so many typos, it is a bit hard for me to evaluate the correctness and significance of the main results.

---

> ### Author Response · Authors · 2021-11-23
> **Response to Reviewer  vUVo**
>
> We sincerely thank the reviewer for the valuable comments and suggestions. Please find our responses in the following.
> ** Missing the related works [1] and [2]**.  We have now added these related works in our revised manuscript. We have also discussed and compared the results in these papers to our results. Due to the problem of typing math formulas in OpenReview please see Section B for details in Supplementary Material.
>
> While both [1] and [2] solve the Bayesian optimization problem which is a non-contextual bandit, our work solves the contextual bandits.
>
> [2] is an unpublished work since 2014 and we believe that the technical analysis in this paper is incorrect.  [1] proposed a solution to avoid this error by using a user-defined parameter $\kappa$ to prevent the posterior variance function $\sigma_{t-1}(.)$ to exceed this lower bound $\kappa$. By this way they can derive an upper bound for their EI algorithm, however, the regret bound depends on $\kappa$ and as $\kappa \rightarrow 0$, this bound quickly explodes and thus, their EI algorithm does not converge.
>
> Compared to [1], our algorithm converges with a sublinear rate for cumulative regret. In addition, while most of EI algorithms are designed to theoretically analyze the simple regret, these two works [1] and [2] focus on analyzing the cumulative regret, however, as discussed in the paragraph above, these analyses are either inaccurate or limited. Thus in our knowledge, our algorithm is the first EI algorithm in the literature that obtains a sublinear cumulative regret guarantee.
>
> **LinEI and NeuralEI modify the original EI by using a threshold function,  whether introducing this threshold function is necessary. Do regret bounds still hold without this modification? If not, can the authors elaborate more on the technical difficulties as well as how this modification solves the issues?**
>
> Thanks for your question! We will explain that a threshold function is necessary and our modifications solve the issues.  Due to the problem of typing math formulas in OpenReview please see our explanation in section Remark on pages 20 and 21 for details in Supplementary Material.  We apologize due to this inconvenience!
>
> **Inflating the posterior variance is somewhat necessary for randomized algorithms (e.g., Thompson sampling) in order to show optimism. I am wondering whether inflating the posterior variance is necessary for deterministic algorithms LinEI and NeuralEI.**
>
> Inflating the posterior variance is necessary to guarantee the correctness of our theoretical analysis. This is due to the existence of the $\beta_t$ which grows over time. $v_t$ is the global scale parameter of the posterior variance and here it plays the role to eliminate the influence of $\beta_t$ so that $\frac{\beta_t}{v_t}$ is bounded as $t$ grows. This technique has been utilized by previous works e.g [Wang and de Freitas., 2014] which analyzes EI in the noisy case, and Bull. [2011] which analyzes EI in the noise-free case. For instance, in [Bull., 2011] that analyses EI under noise-free setting, in section 3.3 they consider the case where the prior can vary with time. The prior at iteration $t$  is a Gaussian process with variance $\sigma^2_t k(.,.)$. Here the role of $v_t$  in our setting is similar to the role of
> $\sigma_t$  in Bull’s work.
>
> **In my opinion, the uniform constant $C'$ in the proof of Lemma 10 depends on $\delta$} (for example, consider $t=1$ )**.  Thanks for correcting us. We have now fixed this error in the proof of Lemma 10. We modified lightly  $v_t = R\sqrt{9d\text{ln}{\frac{t+1}{\delta}}}$  compared to $v_t = R\sqrt{9d\text{ln}{\frac{t}{\delta}}}$ in previous version. With this modification, we can see that the uniform constant $C'$ does not depend on $\delta$. We refer the reviewer to our revised Lemma 10 proof on page xx in the updated manuscript supplementary file.
>
> **I am wondering where Lemma 3 is used in the proof.** We apologize for not clearly mentioning the use of Lemma 3 in our proof of Theorem 1. In our revised submission, we have now clearly mentioned the use of Lemma 3 at the first inequality in the analysis of $R(T)$ at page 7.
>
> **Both $\text{ln}$ and $\text{log}$ are used in the paper**. I think it is consistent to use only one of them.} We corrected it!
>
> **Typos.** We now fixed all typos you mentioned. Thank you very much!

---

### Official Review · Reviewer_UDHJ · 2021-11-09

**Correctness:** 1
**Technical Novelty And Significance:** 3
**Empirical Novelty And Significance:** 2
**Recommendation:** 5
**Confidence:** 4

**Main Review:**


Expected improvement is a popular method to balance exploration and exploitation. While several works analyzed EI in Bayesian optimization and best-arm identification, this paper provides the first analysis of EI in contextual bandits, which I believe is the most significant contribution. The techniques to combine EI with contextual bandits are rather standard, but I do not think this concern is too serious since this topic is novel and important. Still, I would suggest the authors to include more discussions on the technical challenging in combining EI with contextual bandits (e.g., new tools compared with EI for best-arm identification).

Theoretical results, e.g., regret bounds of LinEL and NeuralEI and the claimed $\sqrt{\log(T)}$ regret improvement, are arguably the second contribution of the paper. However, the claim on regret improvement seems incorrect when I check the regret proof of LinEL in detail. I also found many typos / minor errors. The analysis of NeuralEI is analogous to LinEL so similar errors may also exist.

  1. The regret of LinEL should be $O(d\sqrt{T\ln(T)\ln(T^3)\ln(T/\delta)})$. This should be intuitive as when bounding the regret, $\sum_t s_{a(t),t}$ contributes $O(\sqrt{dT\ln(T)})$ following Lemma 3; when $\beta < 3$ we have $\sqrt{2\ln(RT^\beta)} \leq \sqrt{2\ln(RT^3)}$ and $v_t$ contributes another $O(\sqrt{d\ln(T/\delta)})$. Thus the claimed regret improvement does not exists (and it seems that the logarithmic term is worse than OFUL/LinUCB). Please correct me if I was wrong here.

  2. Lemma 10 is not rigorous. When bounding $\beta_t/v_t$, the second equality should be inequality using the fact that $\ln(t^3/\delta) \leq \ln(t^3/\delta^3)$ when $\delta \leq 1$; the third step (inequality) is also not rigorous. Without fixing the issues, $C'$ will depend on $\delta$.

  3. Missing the regularization term $\lambda$. Lemma 1 and Lemma 3 in their original paper used $I$ for initialization instead of $\lambda I$, which was ignored by the authors when applying the lemmas. The authors should revise the proof to reflect $\lambda$.

 Typos:

  - In proof of Lemma 5: "The second one comes from Eq(8)" -> comes from Eq(6); "fifth inequality holds" -> sixth inequality holds;
  - In proof of Lemma 5: "The second equality holds due to the definition of .." is redundant.
  - Combining Lemma 5 and 6: redundant $(t)$ in the second step; missing a matching ']' in the last step
  - In proof of Theorem 3: missing $CC_0/t^{\beta}$ in the second step; summation should be $\sum_t$ instead of $\sum_i$.

========================
In the revised version the authors fixed the errors mentioned above. I am increasing my score to 5.


**Summary Of The Paper:**

This paper applies expected improvement to contextual bandits to balance exploration and exploitation. The authors proposed LinEL for linear rewards and NeuralEI for general rewards using neural network for approximation. Theoretical analysis claimed that compared with UCB based approaches, LinEL's regret bound reduces a factor of $\sqrt{\log(T)}$. Empirical results validated the effectiveness of LinEL and NeuralEI.


**Summary Of The Review:**

Although the theoretical results are flawed, I think most of the errors should be fixable.  I am open to increasing my score if the authors could fix the problems and make the analysis rigorous. The topic of combining EI with contextual bandits is novel and could be interesting to many audiences, even without the claimed regret improvement.

---

> ### Author Response · Authors · 2021-11-23
> **Response to Reviewer UDHJ**
>
> We sincerely thank the reviewer for the valuable comments and suggestions. Please find our responses in the following.
>
> **On the regret of LinEI being $\mathcal O(d\sqrt{T \text{ln}^2(T)\text{ln}\frac{T}{\delta}})$.**
> Thank you very much for correcting us. We have now carefully checked our proof and confirm that the modified regret bound is $\mathcal O(d\sqrt{T \text{ln}^2(T)\text{ln}\frac{T}{\delta}})$ as you mentioned. We would like to note that even with this modification, our regret bound is still sub-linear and matches the lower bound (on the regret of contextual bandits) up to a logarithmic factor. In addition, the obtained regret bound shows that the proposed LinEI can reduce a factor of $\sqrt{d}$ compared to Thompson sampling.
>
> **Lemma 10 is not rigorous. When bounding $\beta_t/v_t$, the second equality should be inequality using the fact that $\text{ln} (t^3/\delta) \le \text{ln} (t^3/\delta^3)$ when $\delta < 1$. The third step (inequality) is also not rigorous. Without fixing the issue $C'$ will depend on $\delta$.**
>
> Thanks for correcting us. We have now fixed this error in the proof of Lemma 10. We modified lightly $v_t = R\sqrt{9d\text{ln}{\frac{t+1}{\delta}}}$ compared to  $v_t = R\sqrt{9d\text{ln}{\frac{t}{\delta}}}$ in previous version. With this modification, we can see that the uniform constant $C'$ does not depend on $\delta$. We refer the reviewer to our revised Lemma 10 proof on page 19 in the updated manuscript supplementary file.
>
> **About the missing the regularization term $\lambda$. Lemma 1 and Lemma 3 in their original paper used $I$ for initialization instead of $\lambda I$, which was ignored by the authors when applying the lemmas.** Thank you to correct us!!
>
> **Typos.** We now fixed all typos you mentioned. Thank you very much!
>
> **I would suggest the authors to include more discussions on the technical challenges in combining EI with contextual bandits (e.g., new tools compared with EI for best-arm identification).** Thank you very much to give us a chance to improve our work. Please see our explanation in the comments to all reviewers.

---

> > ### Comment · Reviewer_UDHJ · 2021-11-29
> > **Thanks for the response**
> >
> > Thanks for the response and for fixing the errors. Current regret and proof look correct to me.
> >
> > Now the regret has an additional $O(\sqrt{\ln T})$ dependency compared with the regret of LinUCB and Thompson Sampling. I suggest the authors revise the writing of the whole paper in the next version to acknowledge this regret difference and add necessary discussion on why EI leads to larger regret than LinUCB and TS (it seems Lemma 6 introduces the additional $O(\sqrt{\ln T})$ and it would be interesting to dig deeper there to see whether it is inevitable). I am increasing my score to 5.

---

### Author Response · Authors · 2021-11-23
**Overall Response**

We thank all reviewers for their insightful comments and suggestions! We have revised the paper to fix the typos and added new experiments.

We would like to address an important question about the  CHALLENGES IN COMBINING EI WITH CONTEXTUAL BANDITS.

First, we show that the "best arm" identification (BAI) problem is not well defined in the setting of the contextual bandit problem.
Let us first consider the setting of a standard MAB problem with $K$ arms. The mean reward of arm $i$, where $i \in [K]$, denoted by $\mu_i$ is unknown but fixed. We assume that $\mu_1 > \mu_2 > ...> \mu_K$  i.e., the arm 1 is the best arm. The goal of the best arm identification is to identify arm 1. This problem is well defined. Next, let us consider a contextual bandit problem with a linear reward function. We note that in this case, the mean reward of arm $i$ depends on a context $x_{i,t}$ which may change at each iteration. It follows that the mean reward for each arm $i$ can be different at different iterations. Thus, the concept of ``best arm'' for contextual bandits does not make sense and the best arm differs depending on context. Therefore, in the setting for context bandits that we consider, we identify an arm at each iteration such that \emph{the cumulative regret is as small as possible}. This means that the goal of our problem is different from that of BAI.

The different goals often reach different algorithms and different techniques. We now show that current techniques of EI in the standard MAB setting are hard to extend to contextual bandits. Indeed, in the standard MAB setting, the EI strategy suggests the arm that offers the maximum ``improvement" over the arm with the largest posterior mean at the current iteration. Therefore, an improvement-based strategy like EI seems suitable for BAI problem whose goal is to seek the arm with the best reward. For example, combining EI strategy with the \emph{allocation matching} which is a \emph{common technique} for BAI, Qin et al. (2017) allocate a sufficient number of samples to the suboptimal arms to eliminate these arms with high confidence. While this is appropriate to solve BAI but it can not be used for contextual bandit problems. This is because it is not possible to eliminate an arm completely as this arm could be the best arm in some context.

To make use of the EI technique to minimize the total regret in contextual bandits, we made a crucial observation that when the largest EI value is "small", by choosing the arm with the largest posterior reward mean instead that with the largest EI value, we can have a tighter regret. This makes our algorithm different from the standard EI algorithm. Next, we address a crucial challenge of how to determine when to use the arm with the largest posterior reward mean instead of that with the largest EI, i.e. to quantify what value of the largest EI value is "small". For this we propose to use a decreasing function of $t$ in the form of $\frac{C_0}{t^{\beta}}$, where $C^0$ is chosen in the interval $[\sqrt{d}, d]$ and $\beta$ is chosen in the interval $[0.5, 3]$ as we explained in the proof of Theorem 1. By this choice, we show that the EI-based strategy can achieve an optimal regret bound like the optimism principle (linUCB).


For the remaining questions, we will respond to each reviewer’s comments individually in the following.

---

### Decision · Program_Chairs · 2022-01-20

**Decision:**

Reject

**Comment:**

Summary: The paper introduces and studies the expected improvement (EI) technique as a way to balance exploration and exploitation for the contextual bandit problem. The authors propose two EI-based algorithms for linear bandits and for neural bandits for a general class of reward functions. The paper presents regret bounds for both methods and shows the experimental results to support their theoretical claims.

Discussion: The reviewers have identified technical issues in the regret bound of LinEL which has been now corrected. Similarly, reviewers have had difficulty assessing the correctness of the paper due to typos and unclear exposition, and raise concerns regarding the amount of corrections that were necessary to reach the current stage. There is no consensus between the reviewers, and some would feel more comfortable if the paper could go through another round of review after major revision.
Reviewer UDHj points that after corrections, "the regret has an additional $O(\sqrt{\ln T})$ dependency compared with the regret of LinUCB and Thompson Sampling." and this should be discussed in the updated version.
Reviewer co1L suggests to compare to the bounds in "The End of Optimism? An Asymptotic Analysis of Finite-Armed Linear Bandits". The authors responded that " To our knowledge, the analysis of the optimal asymptotic regret for contextual bandits as we consider (where the context may be controlled by an adversary) is still an open problem.". In fact, this is the topic of several recent works including:
* "Asymptotically Optimal Information-Directed Sampling" COLT 2021
* "An asymptotically optimal primal-dual incremental algorithm for contextual linear bandits", NeurIPS 2021

The connections of the present work with these two references are strong and should be discussed in more depth. I believe it is a more important discussion than the comparison with the regret bound of LinTS which is yet another problem.

The reviewers have appreciated the originality of the ideas and for that reason we encourage the authors to revise their draft and submit to a future venue.

Recommendation: Reject.